# Early Experience Analyzing Dietary Intake Data from the Canadian Community Health Survey—Nutrition Using the National Cancer Institute (NCI) Method

**DOI:** 10.3390/nu11081908

**Published:** 2019-08-15

**Authors:** Karelyn A. Davis, Alejandro Gonzalez, Lidia Loukine, Cunye Qiao, Alireza Sadeghpour, Michel Vigneault, Kuan Chiao Wang, Dominique Ibañez

**Affiliations:** 1Bureau of Food Surveillance and Science Integration, Health Canada, Ottawa, ON K1A 0K9, Canada; 2Office of Nutrition Policy and Promotion, Health Canada, Ottawa, ON K1A 0L2, Canada

**Keywords:** nutrition, nutrition surveys, usual dietary intake, episodically consumed foods, dietary surveillance, Canadian Community Health Survey, National Cancer Institute method

## Abstract

Background: One of the underpinning elements to support evidence-based decision-making in food and nutrition is the usual dietary intake of a population. It represents the long-run average consumption of a particular dietary component (i.e., food or nutrient). Variations in individual eating habits are observed from day-to-day and between individuals. The National Cancer Institute (NCI) method uses statistical modeling to account for these variations in estimation of usual intakes. This method was originally developed for nutrition survey data in the United States. The main objective of this study was to apply the NCI method in the analysis of Canadian nutrition surveys. Methods: Data from two surveys, the 2004 and 2015 Canadian Community Health Survey—Nutrition were used to estimate usual dietary intake distributions from food sources using the NCI method. The effect of different statistical considerations such as choice of the model, covariates, stratification compared to pooling, and exclusion of outliers were assessed, along with the computational time to convergence. Results: A flowchart to aid in model selection was developed. Different covariates (e.g., age/sex groups, cycle, weekday/weekend of the recall) were used to adjust the estimates of usual intakes. Moreover, larger differences in the ratio of within to between variation for a stratified analysis or a pooled analysis resulted in noticeable differences, particularly in the tails of the distribution of usual intake estimates. Outliers were subsequently removed when the ratio was larger than 10. For an individual age/sex group, the NCI method took 1 h–5 h to obtain results depending on the dietary component. Conclusion: Early experience in using the NCI method with Canadian nutrition surveys data led to the development of a flowchart to facilitate the choice of the NCI model to use. The ability of the NCI method to include covariates permits comparisons between both 2004 and 2015. This study shows that the improper application of pooling and stratification as well as the outlier detection can lead to biased results. This early experience can provide guidance to other researchers and ensures consistency in the analysis of usual dietary intake in the Canadian context.

## 1. Introduction

Health Canada relies on the statistical analysis of dietary surveillance data to help in establishing effective and evidence-based policies, regulations and standard-settings to promote overall health. The 2015 Canadian Community Health Survey—Nutrition [1] (2015 CCHS—See list of Abbreviations in Appendix B) represents the most recent national food consumption dataset. The survey also provides information with respect to nutrient from food and dietary supplements, physical measurements, household food insecurity, and food exclusion (e.g., excluding meat). An important goal of this national nutrition survey is to analyze the usual dietary intake of dietary components (e.g., calcium or fluid milk), by demographic variables of interest, and to estimate the proportion of the population with intakes above or below reference thresholds.

The usual dietary intake represents the long-term consumption of a dietary component; yet population-based nutrition surveys typically use only a few 24-h recalls of individuals to measure dietary variables [2]. These recalls collect information on the quantity and types of foods consumed on a given day. However, since individuals do not generally eat the same foods from one day to another, intra-individual variation is present and must be incorporated into the statistical analysis to avoid over- or under-estimation of nutritional intakes [2]. Hence, statistical methods are used to model usual dietary intake of population to account for the intra- and inter-individual variations present [3].

Since the previous CCHS-Nutrition survey was conducted in 2004 (2004 CCHS), advances in statistical methods for the analysis of usual dietary intake have occurred, and new methods have been proposed [3]. In particular, a joint technical working group comprised of statisticians from Health Canada and Statistics Canada was created in 2016 to evaluate existing statistical methods for estimating the usual dietary intake. After a review of available methods, the working group concluded that the National Cancer Institute (NCI) method [4,5,6] for estimating the usual intake would be recommended for the analysis of the 2015 CCHS data. Many advantages of the NCI method over competing methods were noted, including: Permitting estimation of usual intakes for episodically dietary components (e.g., red meats); permitting adjustment for important covariates of interest; and accounting for the correlation between probability of consumption and amount consumed [5,6,7].

The general reference documentation on the NCI method is available elsewhere [4] including macros for implementation. The NCI method is implemented in three separate collections of SAS macros. The first set models a single dietary component which may be episodically consumed; the second set models two components simultaneously and the third set models many components simultaneously. The focus of this article is to share insights with respect to the first set of macros for modelling a single dietary component.

Objective

The NCI method was originally designed for the National Health and Nutrition Examination Survey (NHANES) [8] conducted in the United States. With a different sampling design, limited advice has been published with respect to implementing the NCI method in the context of CCHS-Nutrition surveys [1,9]. While Statistics Canada has provided a tutorial on “how to” implement the NCI SAS macros in CCHS-Nutrition (available upon request), the objective of this article is to summarize results of early experience implementing the NCI method, and highlight important statistical considerations which arose during the analysis of usual dietary intake. In particular, we considered the following five questions:


*A. Choice of Model:*
Which of the different NCI models should be implemented?
*B. Covariates:*
Which covariates should be included in the model?*C. Stratification* versus *Pooling:*Should age/sex groups be pooled to provide more precise estimates?
*D. Outliers:*
How should outliers be identified and when should they be removed?
*E. Computational Time:*
What is the computational time required by the NCI method?

## 2. Materials and Methods

### 2.1. Data Set and Variables of Interest

The analysis considered dietary consumption data from the 2004 CCHS [10] and the 2015 CCHS [1]—Share files. Both surveys collected detailed information on the consumption of foods, nutrient from foods, and dietary supplements among a representative sample of Canadians living in private households in the 10 provinces. The 2015 CCHS considered respondents aged one and older, whereas the 2004 CCHS sampled all individuals aged zero and older. In both surveys, consumption data were collected using 24-h recalls, in a similar way as the NHANES. Respondents were asked to list all foods and beverages consumed during the 24-h prior to the day of their interview (midnight to midnight), and interviewers used the automated multi-pass method with a five-step approach to help respondents remember their food and beverage choices [11]:(i)A quick list (respondents reported all foods and beverages consumed in whatever order they wished);(ii)questions about specific food categories and frequently forgotten foods;(iii)questions about the time of consumption and type of meal (for example, lunch, dinner);(iv)questions seeking more detailed, precise descriptions of foods and beverages and quantities consumed;(v)a final review of food choices.

A subsample of the population responded to a second 24-h recall 3–7 days later to help assess intra-person variations in the food and beverage intake. The energy and nutrient content of the foods were derived from Health Canada’s Canadian Nutrient File (CNF) Version 2001b Supplement for the 2004 CCHS, and from Health Canada’s CNF 2015 for the 2015 CCHS. In 2004, the initial recall was completed by 35,107 people, and a subsample of 10,786 completed the second recall, yielding response rates of 76.5% and 72.8%, respectively. Correspondingly in 2015, a total of 20,487 people completed the initial 24-h dietary recall, and a subsample of 7608 completed the second recall, yielding response rates of 61.6% and 68.6%, respectively. For this analysis, the CCHS Share files were used, consisting of respondents who agreed to share their data with government analysts for the purpose of research (*n* = 33,469 for 2004 and *n* = 19,673 for 2015). Other important differences between the two nutrition surveys are summarized elsewhere [1,9].

In the present analysis, one nutrient (dietary folate equivalents (DFE)) and one episodically consumed food (red meats) were chosen to represent intakes of daily and episodically consumed dietary components, respectively.

Descriptive statistics for the usual dietary intake (e.g., means, percentiles) and prevalence of inadequacy for DFE (i.e., percent below the estimated average requirement (EAR)) were estimated. Standard errors were computed using bootstrap weights supplied by Statistics Canada [1,10] to properly account for the sampling error in estimation.

Similarly, the means and percentiles of the distribution of usual intakes for other nutrients as well as the comparison of usual intakes to different dietary cut-offs can be found in the Compendium of Nutrient Intakes from Food [12].

### 2.2. Statistical Methods

The NCI method for a single dietary component utilizes two main SAS macros, MIXTRAN and DISTRIB [4]. The MIXTRAN macro fits statistical models to estimate both the probability of consumption and the amount consumed. This is referred to as the two-part model. The probability of consumption may be related to the amount consumed, in which case their correlation can be accounted for in MIXTRAN. For the case where the dietary component is ubiquitously consumed, the probability of consumption is assumed to be one. This case is known as the one-part (or amount-only) model. The DISTRIB macro uses the parameter estimates from MIXTRAN to estimate the distribution of usual intakes as well as the proportion of intakes below or above a certain cut-off [5,6].

For the implantation of the NCI method, a dietary component as well as strata (e.g., age/sex, province) and potential covariates must be identified from the onset. Then, the following questions should be addressed:

*A. Choice of Model:* To determine whether to use the one-part (amount) or two-part (uncorrelated or correlated) model, we considered a similar approach to the one proposed by Krebs-Smith et al. (2010) [2].
If less than 5% of the 24-h recalls (unweighted) within each stratum had zero intake of the dietary component, the amount-only model is used;If greater than or equal to 5% of the 24-h recalls had zero intake, the two-part correlated model is fitted. The correlation between the probability of consumption and the amount consumed is assessed via its Fisher’s transformation using a smaller number of bootstrap replicates (e.g., 50). If it is found to be significant, the correlated model is selected. Otherwise, the uncorrelated model is chosen.

If convergence of the two-part models is not achieved, the amount-only model can be used only if the percentage of the 24-h recalls with zero intakes is between 5%–10%. If more than 10% of the 24-h recalls have zero intakes, the amount-only model over-estimates the usual intake distribution since observations with zero intakes are replaced by one half the minimum observed value. For this case, if the convergence of the two-part model is not achieved, other options should be explored, such as pooling of neighboring age/sex groups or expanding the scope of the dietary component (e.g., from red meats to all meats).

*B. Covariates:* Dietary component intakes vary by age and sex as well as weekday/weekend of the consumption day; these variables were included as covariates in the model. As per NCI recommendations [4], the sequential effect of the recall was considered in the models. Data from 2004 and 2015 CCHS were combined and the survey year was included as a covariate to distinguish between the two surveys and to obtain year-specific estimates. One advantage of including the survey year as a covariate is that it permits hypothesis testing of a difference in the mean usual intake between the two survey years.

Results included in this paper correspond to estimates at the national level. However, for the Compendium of Nutrient Intakes from Food [12], the province is included as a covariate and estimates are generated at the national and provincial levels.

Bootstrapping is used for the estimation of the standard error of covariate estimates. Statistics Canada’s recommendation is to use all of the 500 bootstraps provided with the survey data [1,10]. We tested the impact of the varying number of bootstraps in the size of the standard errors. We compared the use of 50, 100, 200, 300, 400 and 500 replicates.

*C. Stratification* compared to *Pooling:* The NCI method requires sufficient sample size to provide stable estimates of the variance components. This is sometimes difficult to achieve, particularly in the analysis of episodically consumed dietary components. To this end, previous NCI research has considered pooling categories of important covariates (e.g., age groups) so as to borrow information, lead to convergence [2] and generate more precise estimates.

The stratified analysis consists of applying the NCI method on each age-sex group of interest separately. We used the same age-sex groupings recommended by the Institute of Medicine of the National Academies Dietary Reference Intakes (DRI): 1–3 years, 4–8 years (both sexes combined); and separate analysis for males and females aged 9–13 years, 14–18 years, 19–30 years, 31–50 years, 51–70 years and 71 years and older [13].

When pooling the analysis for DFE, we used a similar method proposed by Krebs-Smith et al. (2010) [2], namely children <9 years old; females 9 years and older and males 9 years and older. For red meat, we pooled children <19, females 19+ and males 19+ in order to increase the sample size in the youngest group.

To assess how appropriate the pooling by age/sex groups is as opposed to computing stratified estimates, ratios of within-person to between-person (within-between) variance components were compared. A rule of thumb is that subgroups may be pooled if their estimated within-between variance ratios are similar. Alternatively, the estimated within-between variance ratios obtained from the stratified and from the pooled analysis should be similar under the equal variance assumption.

*D. Outliers:* Outliers were investigated for their impact on statistical estimates. Previous research has indicated that within-between variance ratios larger than 10 may attenuate diet-health relationships in studies with a limited number of dietary recalls [14]. Our proposed method considers domains where the ratio of within-between variance components is greater than 10, and flags observations if they fall more than ±3, ±2.5 or ±2 standard deviations away from the mean for the distribution of the difference between the first and second recalls. With this method, second day recalls of the flagged observation are removed, since second recalls are potentially biased due to the learning effect. The scenario which resulted in the greatest improvement in the within-between variance ratio and removed the fewest observations is selected.

*E. Computational Time:* The computational time associated with each model run was noted. In addition, the possibility of using smaller number of bootstrap replicates to reduce the computational time in different analysis was investigated.

Analysis was completed using the SAS Enterprise Guide 5.1 and the macros MIXTRAN and DISTRIB (Version 2) as available from the NCI website [4]. A significance level of 5% was used throughout.

## 3. Results

We developed a flowchart to help in the application of the NCI method for the case with a single dietary component (Appendix A). This flowchart provides a pathway through the selection of the model to use, the detection of outliers, and the estimation of the distribution of usual intakes.

### 3.1. Choice of Model

Following the choice of a dietary component and of strata of interest, the next step listed on the flowchart is to evaluate the percent of observations with zero intake.

For DFE, among all age/sex groups only 61 observations (less than 1%) had zero intake in either of the recalls, thus the amount-only model was used to estimate usual intakes (Steps 5A–8A of the flowchart).

For red meats, the percentage of zero intakes was greater than 40% for all DRI groups, thus the two-part model was selected (Step 5B). In all situations, the correlation between person-specific effects (Step 6C) was not significantly different from zero (*p* > 0.05). Estimates of the usual intakes were then obtained using the uncorrelated model (Steps 8B–10B).

### 3.2. Covariates

Covariates included in the modelling for DFE and for red meat were: Age/sex groups (when different age/sex groups are pooled), weekend/weekday of recall, and sequential effect of the recall (1st or 2nd recall) as well as survey year, similar to other published recommendations from the National Cancer Institute [4].

Figure 1 illustrates the 2004 and 2015 usual intakes of DFE for (a) females 9–13 and (b) for females 51–70. There is no significance difference between the mean usual intakes of DFE between the two cycles for females 9–13 (*p* = 0.40). Females 51–70 reported significantly larger mean usual consumptions of DFE in 2004 (*p* = 0.002).

Results showed that an increasing number of bootstrap replicates had minimal effect on the significance of the covariates in distinguishing the distributions of DFE from two survey cycles beyond 50 bootstrap replicates. As an example, the *p*-values presented in Figure 1 were calculated using 500 bootstrap replicates. When using only 50 bootstrap replicates instead, the *p*-values for the difference in mean usual intakes by survey year for females 9–13 is unchanged while for females 51–70 it was now found to be *p* = 0.004 as opposed to *p* = 0.002.

### 3.3. Stratification versus Pooling

Usual intakes for DFE and for red meat were obtained using the stratified approach and the pooling of age groups. Key percentiles of the usual intake distribution and estimates of the ratio of within-between variance for DFE and red meats for the 2015 CCHS data are presented in Table 1 and Table 2 respectively. Note that only results for children and females are presented. Results for males and for the 2004 CCHS were analogous.

When comparing estimates between the stratified and pooled approaches, we considered the relative difference in the mean and percentile estimates as well as the ratio of within-between variance. In particular for DFE intakes, we noted that the estimates of the mean and median were similar for almost all age/sex groups in the pooled or stratified analysis (differing by less than 4%, except for females 9–13) (Table 1). However, estimates of the 5th percentile differed by 17.3% (children 1–3 years) and 27.5% (females 9–13 years) between the pooled and stratified results. Similarly, large differences were noted at the 95th percentile (9% and 9.4% difference, respectively). Figure 2a illustrates the larger tails associated with the usual intake distribution for the pooled analysis over the stratified analysis for females 9–13 years. Figure 2b shows that the two methods yielded similar distributions for females 19–30. For age groups in which large differences were found, we noted the within to between ratio was much different for the stratified versus the pooled analysis (Table 1), indicating that the equal variance assumption may not hold and pooling these groups results in different estimates of usual intake distributions.

For red meats, original stratification yielded a within-between variance ratio >10 for children 2–3 even after excluding outliers (data not shown). As a result, the smallest strata where the within-between ratio was <10 was when children 2–3 years old were pooled with children 4–8. In Table 2, results listed for children 2–3 and for children 4–8 under “stratified” were obtained from pooling these age groups together. Listed under “pooled” for the same age groups are the results obtained when pooling children <19.

A similar situation occurred with females 19–30 where the initial within-between variance ratio was >10. In Table 2, results listed for females 19–30 and females 31–50 under “stratified” were obtained from pooling these two age/sex groups together. Listed under “pooled” for the same age groups are the results obtained when pooling females 19+.

Similar to DFE, with the exception of the 5th percentile, the largest differences were found for those with greater differences in the ratio of within/between variances (female 14–18 years and female 71+). Interpretation with caution is advised in all 5th percentiles reported due to the large variability of the estimates (i.e., CV in [16.6, 33.3]). We also compared the impact of using 250 and 500 bootstraps replicates. Table 3 provides estimates of standard errors obtained using the pooled and stratified approaches for DFE, using 250 and 500 bootstrap replicates. Standard errors from the stratified analysis were equivalent or larger than those from the pooled analysis. The pooled estimates are expected to be more precise as they are based on a larger sample and “borrow strength” from other observations to better estimate the variance components. When increasing the number of bootstrap replicates, minimal improvement to standard error estimates were found when more than 250 bootstrap replicates were used. This result is consistent with other published literature [6].

### 3.4. Outliers

For DFE, ratios of the within-between variance were less than 10 for all DRI groups except children 4–8. For this group, 17 observations were flagged as outliers and their second day of recall were removed. For red meats, no outliers were removed for the pooled analysis, whereas for the stratified analysis, 109 out of 6396 s recalls were removed over five DRI groups (9 for children 2–3; 32 for children 4–8; 23 for females 9–13; 21 for females 31–50; and 24 for females 51–70).

### 3.5. Computational Time

As expected, the two-part model requires more time to reach convergence as compared to the amount-only model. Pooled analysis requires more time as compared to the total computation time of all the strata analyzed separately. Based on our analysis, the computational time for the stratified analysis was found to be between 2–5 h per DRI age/sex group for red meat and 1–2 h for DFE. Considering the pooled analysis, there was a 50% increase in the total computational times of the two-part model over the one-part model. Additionally, the total time for DFE was 23 h for the stratified analysis and 44.4 h for the pooled analysis. For red meats, the corresponding total computational times were 48.2 h and 66.1 h, respectively.

## 4. Discussion

While various authors have analyzed usual dietary intakes from the CCHS-Nutrition survey using the NCI method [15,16,17,18,19], few authors have assessed the statistical considerations mentioned in the present article. The NCI method applied to the CCHS-Nutrition data offers computational advantages for estimation of usual dietary intake such as the ability to analyze episodically consumed foods (and nutrients) and the inclusion of covariates. However, increased analytical capability comes with an increased number of statistical assumptions, which should be verified as part of the analysis. In particular, statistical considerations such as the assumption of equal variance, choice of model, correlation of variance components and the presence of outliers should be acknowledged as part of an analysis of usual intakes using the NCI method.

We propose the following recommendations when analyzing dietary intake data from the 2004 and 2015 CCHS using the NCI method:Consider the unweighted proportion of the 24-h recalls having zero intake of a nutrient or food in order to determine whether a one- or two-part model should be used.If different categorical covariates are of interest (e.g., DRI age/sex groups; income levels), use the MIXTRAN macro to compute estimates of the parameters in two ways: (a) Pooling all levels by using a covariate and (b) separate analysis for every level of the covariate. In each case, compare the ratio of the within-between variance components, and consider a stratified analysis if this variance ratio is at least 30% higher or lower than for the other approach.When analyzing dietary components that are rarely consumed (i.e., very episodic), there might not be enough respondents with non-zero intake to obtain stable estimates of the variance components. In addition, estimated distributions may show large variability (i.e., large CVs), particularly in the tails of the distribution. For such cases, a recommendation would be to retain broader nutrient and/or covariate categories. For example, an analyst could model all meats instead of red meats; or pooling with other age/sex groups is a viable option, although not always recommended.Perform outlier detection by evaluating recalls which may affect equal variance assumptions. Remove outliers and re-fit using the MIXTRAN macro to evaluate the impact on parameter estimates.

Appendix A presents a flowchart for the univariate analysis of usual intakes using the NCI method, based on results from the present article.

Future research would include evaluation of the bivariate and multivariate NCI macros as applied to the CCHS-Nutrition surveys, which are available for more complex scenarios involving ratios of two or more dietary components [20,21,22].

Despite the many advantages of the NCI method, one difficulty is that the method requires a large sample size in order to generate reliable estimates, which in turn may necessitate pooling subgroups. This issue intensifies for episodically consumed dietary components. However, a consequence of pooling groups is that the pooled variance components may incorrectly estimate the variances within subgroups when the equal variance assumption does not hold. Consequently, the adjustment factor will be inaccurate, leading to bias in the estimation of the usual intake distribution. The other caveat is the decision of how to pool. In this research, the analyses were pooled by neighboring age/sex groups. However, pooling may be performed using other groups or variables, for example neighboring provinces. Choosing the type of pooling strategy is a balance between the adequate sample size, the equal variance assumption, and the appropriate research question.

Retaining observations flagged as outliers in the analysis may result in over-estimation of the ratio of the within-between variance. This, in turn, inflates the estimated standard errors to a point that the estimates are no longer publishable according to Statistics Canada’s guidelines (i.e., large CV’s). In this research, we proposed a method which removes outliers based on the difference between the two days of recall. Another approach recommended by Krebs-Smith et al. (2010) [2] is to Box-Cox transform the raw non-zero values and flag observations which were 2.5 times the interquartile range below the 25th and above the 75th percentiles, respectively. Our proposed method identifies outliers as those points having the greatest impact on the within-between variation whereas the second method characterizes observations which may violate normality assumptions.

Lastly, implementation of the NCI method for large surveys such as CCHS that requires bootstrapping is computationally intensive and may limit the ability to perform standard model diagnostics. We noted that increasing the number of bootstraps has minimal effect on the significance of the covariates or the correlation coefficient. To reduce the computational time, we recommend employing model building steps with fewer bootstraps weights (e.g., 50). After finalizing the model, results may be run for all the 500 bootstraps replicates to have precise estimates of standard errors.

## 5. Conclusions

The use of the NCI method is a valuable tool for Canadian nutrition surveys. Lessons learned from early experience using the NCI method lead us to the development of a flowchart to facilitate the choice of the NCI model to use. The ability of the NCI method to include covariates permits comparisons between both the 2004 and 2015 CCHS and informs health policymakers and regulators as to changes in the prevalence of nutrient inadequacy over time. This study shows that the improper application of pooling and stratification as well as outlier detection can lead to biased results. This early experience can provide guidance to other researchers and ensures consistency in analysis of usual dietary intake in the Canadian context.

## Figures and Tables

**Figure 1 nutrients-11-01908-f001:**
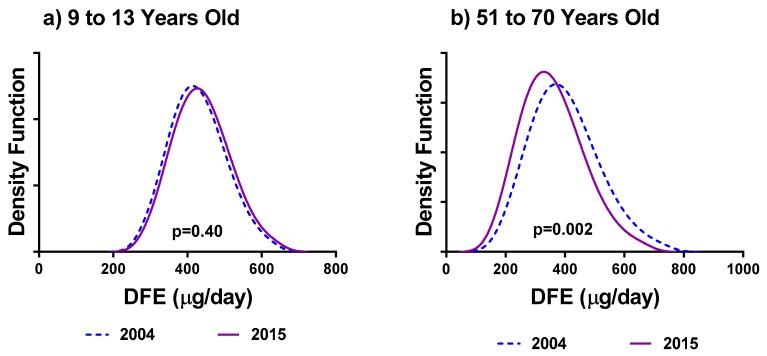
Comparison of estimates of usual intake distribution of dietary folate equivalents (DFE) (μg/day) between 2004 and 2015 for (**a**) females 9–13 years old; and (**b**) females 51–70 years old.

**Figure 2 nutrients-11-01908-f002:**
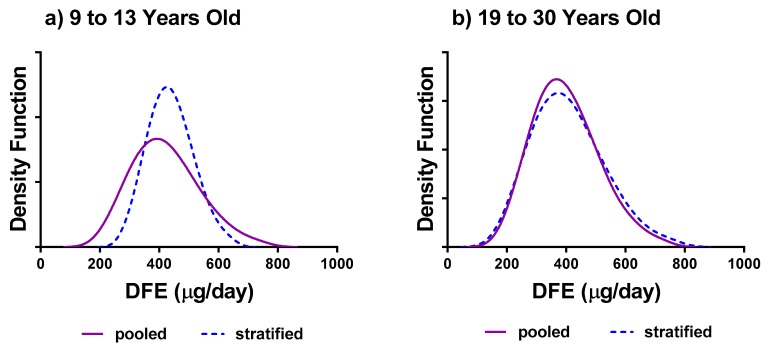
Comparison of estimates of usual intake distribution of DFE (μg/day) between pooled versus the stratified approach for (**a**) females 9–13 years old; and (**b**) females 19–30 years old. Data Sources: Statistics Canada, 2004 and 2015 Canadian Community Health Survey—Nutrition Share files.

**Table 1 nutrients-11-01908-t001:** Comparison of distributional estimates of dietary folate equivalents (μg/day) by the National Academies Dietary Reference Intakes (DRI) groups—National Cancer Institute (NCI) Method—Amount Model.

DRI Age/Sex Group	Method	Ratio of Within/Between	2015 Usual Intake Distribution					
			Mean	Percentiles				% < EAR
				5th	50th	90th	95th	
Children 1–3 years	Pooled	3.6	281.7	174.1	275.8	377.2	408.5	<3
	Stratified	1.6	281	144	273.4	403.8	445.4	F
	***% Diff*^1^**		***0.2***	***17.3***	***0.9***	***7.1***	***9***	
Children 4–8 years	Pooled	3.6	401.1	265.2	394.6	520.2	560.4	<3
	Stratified	5.5	398.8	284.3	392.9	500.1	533.9	0
	***% Diff***		***0.6***	***7.2***	***0.4***	***3.9***	***4.7***	
Females 9–13 years	Pooled	2.5	421.9	249.9	410.1	578.7	634.2	5.01 ^E^
	Stratified	6.5	438.2	318.5	433.5	540.4	574.3	<3
	***% Diff***		***3.9***	***27.5***	***5.7***	***6.6***	***9.4***	
Females 14–18 years	Pooled	2.5	412.7	244.9	400.8	564.8	620.1	24.87
	Stratified	1.9	412.2	219.1	398.9	588.5	649.6	28.41
	***% Diff***		***0.1***	***10.5***	***0.5***	***4.2***	***4.8***	
Females 19–30 years	Pooled	2.5	399.6	233.4	387.4	550.4	607.3	25.95
	Stratified	2.1	407.3	226.7	394.4	572.5	632.3	25.87
	***% Diff***		***1.9***	***2.9***	***1.8***	***4***	***4.1***	
Females 31–50 years	Pooled	2.5	401	234.9	389.5	551.2	604.1	25.11
	Stratified	3.5	407.5	255.8	397.8	545.2	593.6	20.37 ^E^
	***% Diff***		***1.6***	***8.9***	***2.1***	***1.1***	***1.7***	
Females 51–70 years	Pooled	2.5	369.8	214	358.2	512.6	563.5	35.05
	Stratified	1.9	358.9	201.9	347.7	502.3	555.5	39.55
	***% Diff***		***2.9***	***5.7***	***2.9***	***2***	***1.4***	
Female 71+ years	Pooled	2.5	339.7	193.1	329.3	473.1	522.7	46.29
	Stratified	1.8	332.2	193.7	323.3	458	501.2	48.52
	***% Diff***		***2.2***	***0.3***	***1.8***	***3.2***	***4.1***	

Data Sources: Statistics Canada, 2004 and 2015 Canadian Community Health Survey—Nutrition Share files. Symbol Legend: E Data with a coefficient of variation (CV) from 16.6% to 33.3%; interpret with caution; F Data with a coefficient of variation (CV) greater than 33.3% with a 95% confidence interval not entirely between 0 and 3%; suppressed due to extreme sampling variability; <3 Data with a coefficient of variation (CV) greater than 33.3% with a 95% confidence interval entirely between 0 and 3%; interpret with caution; ^1^ Absolute percent difference relative to Pooled Method computed for means and percentiles (e.g., $\frac{\left|\left(\mathrm{Pooled}\;\mathrm{Mean}-\mathrm{Stratified}\;\mathrm{Mean}\right)\right|*100}{\mathrm{Pooled}\;\mathrm{Mean}}$). EAR, estimated average requirement.

**Table 2 nutrients-11-01908-t002:** Comparison of distributional estimates of red meats (gram) by DRI groups—NCI Method —Two Part Model—Uncorrelated.

DRI Age/Sex Group	Method	Ratio of Within/Between	Mean	Percentiles			
				5th	50th	90th	95th
Children 2–3 years	Pooled ^1^	8.1	17.5	5.6 ^E^	15.7	30.3	35.4
	Stratified ^2^	7.4	16.3	4.2 ^E^	14.6	29.1	34.4
	***% Diff*^4^**		***6.7***	***24.2***	***7.1***	***3.7***	***2.9***
Children 4–8 years	Pooled	8.1	24.5	8.2 ^E^	22.2	41.3	48
	Stratified ^2^	7.4	22.7	6.36 ^E^	20.7	39.5	46.1
	***% Diff***		***7.4***	***23.9***	***7.2***	***4.5***	***4***
Females 9–13 years	Pooled	8.1	27.7	9.5^E^	25.5	46.3	53.3
	Stratified	6.1	29.9	F	27.1	51.6	60.3 ^E^
	***% Diff***		***7.6***	***F***	***6***	***11.4***	***13.1***
Females 14–18 years	Pooled	8.1	28.7	9.3 ^E^	26	49.1	57.2
	Stratified	3	23.8	F	20.1	44.9	54.7
	***% Diff***		***17.2***	***F***	***22.9***	***8.6***	***4.3***
Females 19–30 years	Pooled	7.3	31.6	9.5 ^E^	28.8	54.7	63.5
	Stratified ^3^	7.8	32.4	8.0 ^E^	29	58.6	68.5
	***% Diff***		***2.5***	***15.4***	***0.7***	***7.1***	***7.8***
Females 31–50 years	Pooled	7.3	35.3	11.1 ^E^	32.4	59.9	69.1
	Stratified ^3^	7.8	35.9	9.3 ^E^	32.8	63.3	73.2
	***% Diff***		***1.6***	***16.6***	***1.2***	***5.5***	***5.9***
Females 51–70 years	Pooled	7.3	38.5	12.5 ^E^	35.6	64.6	74.6
	Stratified	5.5	36.8	16.4 ^E^	34.5	57.1	65.1
	***% Diff***		***4.5***	***30.8***	***3.2***	***11.6***	***12.7***
Females 71+ years	Pooled	7.3	34.4	10.9 ^E^	31.8	58.1	67
	Stratified	3	31.3	F	27.2	58.3	69.9
	***% Diff***		***9.2***	***F***	***14.4***	***0.3***	***4.3***

Data Sources: Statistics Canada, 2004 and 2015 Canadian Community Health Survey—Nutrition Share files. Symbol Legend: E Data with a coefficient of variation (CV) from 16.6% to 33.3%; interpret with caution; F Data with a coefficient of variation (CV) greater than 33.3%; suppressed due to extreme sampling variability; ^1^ Pooling strategy: Children and adolescents < 19; Males 19+ and Females 19+; ^2^ For children 2–3, a stratified analysis yielded a within-between variance ratio >10. So, a pooled estimate of 2–3 and 4–8 years is provided; ^3^ For females 19–30, a stratified analysis yielded a within-between variance ratio >10. So, a pooled estimate of females 19–30 and 19–50 is provided; ^4^ Absolute percent difference relative to pooling strategy computed for means and percentiles (e.g., $\frac{\left|\left(\mathrm{Pooled}\;\mathrm{Mean}-\mathrm{Stratified}\;\mathrm{Mean}\right)\right|*100}{\mathrm{Pooled}\;\mathrm{Mean}}$).

**Table 3 nutrients-11-01908-t003:** Comparison of standard errors (SE) of distributional estimates of dietary folate equivalents (μg/day)—Canadian Community Health Survey (CCHS) 2015—Nutrition.

DRI Age/Sex Group	No. of Bootstrap Replicates	Pooled			Stratified		
		SE of Mean	SE of 5th Percentile	SE of 95th Percentile	SE of Mean	SE of 5th Percentile	SE of 95th Percentile
Children 1–3 years	250	6.9	14.2	17.5	8.5	12.5	16.3
	500	6.5	14.6	18.5	8.1	12.3	17
Children 4–8 years	250	7.8	15.9	24.1	8.7	24.2	35.5
	500	7.9	16.7	25.3	8.9	24.6	35
Females 9–13 years	250	6.6	8.1	14.4	10.3	21.6	27.7
	500	6.8	8.3	14.6	10.2	22.2	28.2
Females 14–18 years	250	7.4	8.4	15.3	*11.9*	*15*	28.7
	500	7.7	8.5	15.7	*11.7*	*14.5*	29.4
Females 19–30 years	250	9.3	8.8	18.3	13.8	16.1	36.1
	500	8.8	8.4	17.8	13.6	15.9	34.9
Females 31–50 years	250	6.9	8.1	14.1	10.2	20.9	35
	500	6.7	8.1	14	9.9	20.8	34.7
Females 51–70 years	250	4.7	7.3	11.1	6.3	11.4	18.2
	500	4.9	7.5	11.2	6.3	11.5	17.9
Females 71+ years	250	4.3	6.7	10.7	5.7	8.4	15.3
	500	4.6	6.7	11.1	6	8.5	14.7

Data Sources: Statistics Canada, 2004 and 2015 Canadian Community Health Survey—Nutrition Share files.

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
