# Peer review of "Early Experience Analyzing Dietary Intake Data from the Canadian Community Health Survey—Nutrition Using the National Cancer Institute (NCI) Method"

_nutrients, 2019, doi:10.3390/nu11081908_

Round 1
Reviewer 1 Report
Review Nutrients, ID: nutrients-498780
Title: Early experience analyzing dietary intake data from the Canadian Community Health Survey - Nutrition using the National Cancer Institute (NCI) method
Overall
The article is on aspects of analyzing intake data, which is an always interesting as well as important topic.
Overall I find the article interesting and well written, however readability could be improved. Regard my below comments as suggestions for making the text more accessible for the readers.
The text contains many abbreviations and for readers not fully into the area they are hard to remember and relate to although the letter combinations are explained in the text. I suggest a list of abbreviations to make the reading easier.
Abstract
The abstract is good, but may be improved by less details. Make it shorter and draw the big lines.
Introduction
Very good. May improve the whole paper by including a list of abbreviations (se above)
Objective
Focus more on the objective. Also the research questions should be in this paragraph – now they are written in the Material and Methods-paragraph. Ie questions regarding Choice of models, Covariates, Pooling vs Stratification, Outliers and Computation time.
Material and Methods
See above regarding research questions. The choice of macros (MIXTRAN and DISTRIB) might be described.
Results
This paragraph is extremely hard to read if the reader do not keep track of all abbreviations, a list would be very helpful. Check tables so they are all get the same fonts and text sizes.
Discussion
The text is good, but too short. Please add more to make the reader understand. Also a discussion of more results in comparison to more references would be interesting.
Conclusion
The reference to updating a compendium on intake does not belong to the conclusion and should be removed to the discussion. If using suggested research questions, these should be answered in the conclusion.

Reviewer 2 Report
An interesting paper, which is well written, but perhaps has limited audience value.
I have a few queries:
Please try to put each table on a separate page with headings if it overlaps on to two pages. Some of the larger tables might be presented in landscape.
It would be interesting to know how the amounts of folates per group meet the RDI. I know its not the remit of the paper but would be of interest to people using the study.
I am concerned with the addition of two data sets (2004 and 2015) to meet the sample size. Surely this would add other risks of bias - hiding differences in eating patterns between the years??
